# Metabolomics of *Solanum lycopersicum* Infected with *Phytophthora infestans* Leads to Early Detection of Late Blight in Asymptomatic Plants

**DOI:** 10.3390/molecules23123330

**Published:** 2018-12-15

**Authors:** Paula Galeano Garcia, Fábio Neves dos Santos, Samantha Zanotta, Marcos Nogueira Eberlin, Chiara Carazzone

**Affiliations:** 1Laboratory of Advanced Analytical Techniques in Natural Products, Universidad de los Andes, Bogotá 111711, Colombia; plgaleanog@gmail.com; 2Bioprospección de los Productos Naturales Amazónicos, Facultad de Ciencias Básicas, Universidad de la Amazonia, Florencia 180002, Colombia; 3ThoMSon Mass Spectrometry Laboratory, University of Campinas, Institute of Chemistry, Campinas 13083-970, Brazil; fabiof6@gmail.com (F.N.d.S.); mneberlin@gmail.com (M.N.E.); 4Laboratório de Diagnostico Fitopatológico, Instituto Biológico, São Paulo 04014-900, Brazil; sa_zanotta@terra.com.br

**Keywords:** late blight, *Phytophthora infestans*, LC-MS, MALDI-MS, multivariate analysis, plant-pathogen interaction, tomato

## Abstract

Tomato crops suffer attacks of various pathogens that cause large production losses. Late blight caused by *Phytophthora infestans* is a devastating disease in tomatoes because of its difficultly to control. Here, we applied metabolomics based on liquid chromatography–mass spectrometry (LC-MS) and metabolic profiling by matrix-assisted laser desorption ionization mass spectrometry (MALDI-MS) in combination with multivariate data analysis in the early detection of late blight on asymptomatic tomato plants and to discriminate infection times of 4, 12, 24, 36, 48, 60, 72 and 96 h after inoculation (hpi). MALDI-MS and LC-MS profiles of metabolites combined with multivariate data analysis are able to detect early-late blight-infected tomato plants, and metabolomics based on LC-MS discriminates infection times in asymptomatic plants. We found the metabolite tomatidine as an important biomarker of infection, saponins as early infection metabolite markers and isocoumarin as early and late asymptomatic infection marker along the post infection time. MALDI-MS and LC-MS analysis can therefore be used as a rapid and effective method for the early detection of late blight-infected tomato plants, offering a suitable tool to guide the correct management and application of sanitary defense approaches. LC-MS analysis also appears to be a suitable tool for identifying major metabolites of asymptomatic late blight-infected tomato plants.

## 1. Introduction

Tomato (*Solanum lycopersicum*) is one of the most widespread vegetables worldwide, with more than 160 million tons produced in 2013 alone [1]. Tomatoes are highly consumed, mainly as fresh fruits or processed products, due to their nutritional and beneficial properties to human health [2,3,4]. These crops, however, suffer attacks of various pathogens such, as viruses, bacteria, fungi and nematodes, causing substantial production losses. Late blight, caused by the phytopathogenic oomycete *Phytophthora infestans*, is one of the most devastating tomato diseases, demanding high chemical input for disease control worldwide [5,6,7,8]. On farms with conditions favorable to pathogens, such as a high humidity and temperature, late blight can cause severe epidemics and destroy the entire tomato crop production [9]. The expenses due to tomato late blight and control measures are estimated to exceed $5 billion annually worldwide [10]. The development of resistant cultivars is an important alternative for pathogen management and control, but as yet there are no highly resistant cultivars. The early detection of asymptomatic infected *S. lycopersicum* with *P. infestans* is therefore fundamental to establish effective strategies for pathogen management and control. 

*P. infestans* exhibits a two-phases life cycle. Post infection in the host is characterized as an asymptomatic biotrophic phase, and a late necrotrophic stage is characterized by tissue degradation and disease [11,12]. The identification of *P. infestans* is realized by traditional techniques that require isolation from the plant tissue followed by culture-based morphological evaluation, but this protocol requires a high level of taxonomic expertise [13]. Recently, new techniques of plant pathogen diagnosis, including monoclonal antibodies, enzyme-linked immune sorbent assay (ELISA) and DNA-based techniques, have been introduced, which are far more specific, sensitive and accurate [14,15,16,17] than the traditional techniques. These techniques efficiently identify the pathogens, but they fail to provide information on the pathogen-host interaction at the molecular level or the biochemical alterations caused by late blight. 

Mass spectrometry techniques are known to provide rapid and accurate target and untargeted analysis of metabolites, allowing the detection of pathogens in plants and detection of the metabolites of the pathogen-host interaction. MALDI-MS has been applied to evaluate metabolic alterations due to different phytopathogen infestations. MALDI-MS was able to establish metabolic interactions between rice−bacterium (*Oryza sativa* infected with *Xanthomonas oryzae*) and soybean−aphid (*Glycine max* colonized with *Aphis glycines*) [18]. Additionally, MALDI-MS profiles of proteins, lipids and metabolites have also revealed plant-pathogen interactions [14,19,20], specifically, the protein profile of sugarcane after infection by *Sporisorium scitamineum*, [21] the identification of differential proteins of rice leaves infected with the fungus *Cochliobolus miyabeanus* [22], and the proteins that may lead to the resistance of tomato plants to *P. infestans* [7].

Metabolomics based on LC-MS and GC-MS approaches reveal changes in the metabolic profiles of the plants as a response to the attack of pathogens [11,23,24,25,26,27,28,29]. Metabolomics has been applied to investigate the biochemical responses of tomato to the attacks of single and multiple pathogen infestations; the discrimination of different times post infection; the differentiation between non-infected and infected plants; the susceptibility/resistance to several pathogens; and even the beneficial interactions with mycorrhizal fungi [30]. All of these approaches have provided a deeper insight into biological processes and supported the discovery of potential biomarkers [29,31,32]. The metabolomics studies of infected tomatoes have focused on the identification of metabolic pathways and those metabolites that are up- and downregulated by infection with tomato mosaic virus (ToMV) [33]; tomato yellow leaf curl virus (TYLCV) [34]; potato spindle tuber viroid (PSTVd) [35]; root-knot nematode (RKN) *Meloidogyne incognita* [36]; *Pseudomonas syringae* and *Botrytis cinerea* [37], and infestation with spider mites (*Tetranychus urticae*) and aphids (*Myzus persicae*) [38,39]. There are, however, only a few metabolomics studies about the metabolic interactions between tomato cultivars and *P. infestans* that have focused on the early detection of infected plants. Here, we applied metabolomic analysis of LC-MS and MALDI-MS profiles associated with multivariate data analysis in the early detection of tomato infection by *P. infestans* and differentiated between the infection times.

## 2. Results and Discussion

### 2.1. Multivariate Statistical Analysis of LC-MS Metabolomics Data

To investigate the comprehensive metabolic changes that occur in response to infection by *P. infestans*, tomato plants (Santa Cruz Kada cultivar) were inoculated with a sporangial solution of *P. infestans* and collected at 4, 12, 24, 36, 48, 60, 72 and 96 h post inoculation (hpi). At 96 hpi, the infected asymptomatic leaves were collected and examined under optical and stereoscopic microscope to confirm the sporangia structures of *P. infestans*. In the LC-MS untargeted metabolomics analysis, the ion chromatograms obtained in the ESI positive and negative ion modes (see Appendix A, respectively) showed no apparent differences between the infected and non-infected samples, but the Cloud Plot univariate analysis showed 5582 and 4790 features for the positive and negative ion modes, respectively, with statistical significance (ANOVA *p* ≤ 0.01, see Appendix A). 

In the first step of the statistical processing, the principal component analysis (PCA), a technique used for dimensionality reduction of multivariate data whilst preserving most of the variance [40], was applied to LC-MS metabolic profiles data in order to find cluster of samples. The PCA score plot (Appendix A) clearly shows three clusters of the samples according to the different infection times, corresponding to early asymptomatic infection (4, 12, 24 and 36 hpi), late asymptomatic infection (48, 60, 72 and 96 hpi) and the initial control. 

Secondly, multivariate analysis of OPLS-DA was then applied, and it revealed significant differences between the control and early- and late-infected asymptomatic plants, along with additional information regarding significant molecular features of the infection stages. In the OPLS-DA plots (Figure 1a,b), three different clusters in both the positive and negative ion modes corresponding to the initial control, early infection (4 to 36 hpi, cluster I) and late infection (48 to 96 hpi, cluster II) were delineated. Additionally, the heat map plots based on HCA showed additional information about the similarities between samples and clusters [41]. 

The variable importance in projection (VIP) scores, which estimate the importance of each variable in the projection used in a PLS model [42], show potential discriminant metabolites with high score values and high discriminator power (Figure 1c,d). The annotated metabolites isocoumarin (M301T17); the diterpene lactone (M579T17) (Figure 1c) and the triterpene saponin (M984T24) (Figure 1d) increase in intensity with the progression of the infection, whereas the intensity of the triterpenoid (M685T19); the peonidin 3-(4-sinapoylgentiobioside) (M832T22) (Figure 1c) and the sulfoquinovosyldiacylglycerol (M820T23) (Figure 1d), decrease. The level of annotation and additional information on these metabolites are summarized in Appendix A.

Subsequently, partial least squares-discriminant analysis (PLS-DA) that is a supervised chemometric method used to optimize the separation between different groups [40], was applied to investigate specific metabolic changes in the early asymptomatic infection stage. In Appendix A, the differentiation of the infection times (4, 12, 24 and 36 hpi) based on metabolic profiles is now apparent. The VIP score-plots derived from PLS-DA (Figure 2a,b) can also discriminate other possible metabolites, such as the triterpene saponin (M798T23), the demissine (M529T12) (Figure 2a) and the triterpenoid saponin (M791T21) (Figure 2b), which increase at the early asymptomatic infection stage. For level of annotation see the Appendix A.

Similarly, PLS-DA analysis was applied to investigate the specific metabolic changes at the late infection stage (48 to 96 hpi, cluster II). The differentiation of the 48, 60, 72 and 96 hpi is now also apparent. The PLS-DA plots of Appendix A confirm the closeness between the post inoculation time points of the cluster II. The gradient variation of the component between 48 and 96 for the LC-MS data in both positive and negative ion modes is notable. 

The VIP plot (VIP > 5) of the late infection stage allows the identification of late discriminant metabolites. The polyacetylene (M277T16); the isoprene (M367T19); the carboxylic acid esters (M207T14) and the macrolide (M253T13) (Figure 3b) features show a gradual increase in relative abundance from 48 to 96 hpi. Other features are also present in the s-plot (Appendix A), supporting the idea that the bulk of the potential biomarkers increase as a function of infection time.

These discriminant molecular features of the Figure 2a,b and Figure 3a,b are in concordance with the ions depicted in the s-plot derived from the OPLS-DA model (Appendix A), where the p(corr) axis represents the reliability of each variable and has a value of between + 1 and − 1. Variables in the extreme lower left and upper right quadrants are reliable for the extraction of putative biomarkers [28]. The permutation test applied to the LC-ESI (+)-MS data from the early and late asymptomatic infection stages (4–96 hpi and control) revealed a Q2 value of 0.691 and an R2Y of 0.995 (*p* < 5 × 10^−4^) (Appendix A). Likewise, the LC-ESI (−)-MS data from the early and late asymptomatic infection stages (4–96 hpi and control) revealed a Q2 value of 0.893 and an R2Y of 0.977 (*p* < 0.003) (Appendix A). The permutation test applied to the multivariate LC-ESI (+)-MS data from only at the early asymptomatic infection stage (4–36 hpi) revealed a Q2 value of 0.85 and an R2Y of 0.999 (*p* < 5 × 10^−4^), and multivariate LC-ESI (+)-MS data from only at the late asymptomatic infection stage (48–96 hpi) revealed a Q2 value of 0.741 and an R2Y of 0.998 (*p* < 5 × 10^−4^). Similarly, the LC-ESI (−)-MS data (4–36 hpi) revealed a Q2 value of 0.943 and an R2Y of 0.997 (*p* < 0.01), and the LC-ESI (−)-MS data (48–96 hpi) revealed a Q2 value of 0.908 and an R2Y of 0.984 (*p* < 0.01). These results indicate that the OPLS-DA model has good fitness and prediction power. Cross validation (CV) of the PLS-DA models by leave-one-out (LOO) or 10-fold revealed five latent variables (components) for the optimal performance of the multivariate data (Appendix A).

The discriminant molecular features (Table 1) M845T24 (*m*/*z* 845.4816), M984T24 (*m*/*z* 983.5392) and M798T23 (*m*/*z* 797.5049) were annotated as triterpene saponins using MS and MS/MS spectra in the MS-Finder database. Similarly, the features M529T12 (*m*/*z* 528.7643) and M737T20 (*m*/*z* 737.4186) were annotated as the steroidal saponins demisine and tuberoside J, respectively. The triterpene and steroidal saponins are usually reported to have important roles in the defense response of plants against pathogens, pests and herbivores due to their antimicrobial, antifungal, antiparasitic, insecticidal and anti-feed properties [43,44,45,46]. The saponins are produced by tomatoes and have been studied in detail in relation to their potential role in the defense response of plants against phytopathogenic fungi [45,47]. The biosynthesis of triterpenoid saponins is induced in the roots in response to *Phytophthora cactorum* attack in roots of *Fragaria vesca* [48]. 

Using the MS and MS/MS spectra obtained from the LC-MS analysis, the following compounds were successfully annotated: steroidal glycoalkaloids α-tomatine (M1079T12; *m*/*z* 1078.5436), dehydrotomatine (M1077T12; *m*/*z* 1076.527), hydroxytomatine isomer I (M1095T10; *m*/*z* 1094.5382), the unknown glycoalkaloids UGA 11 (M1109T12; *m*/*z* 1108.5541) and UGA 28 (M1121T12; *m*/*z* 1120.5120) and the aglycon tomatidine (M414T20; *m*/*z* 414.3385) (Table 1), which occur naturally in tomatoes [49,50,51]. However, some studies of glycoalkaloids have shown that these molecules have antibiotic properties against a variety of fungi [52,53], suggesting that tomatine (α-tomatine and dehydrotomatine) may play a major role in disease resistance in the tomato plants [49,51,54]. Tomatidine is an important biomarker of infection, because bacterial and fungal pathogens secrete various types of tomatinase enzymes that can detoxify α-tomatine by removing one or more sugar residue [55,56]. Thus, it has also been suggested that products resulting from tomatinase activity play an indirect role in the virulence of pathogens against tomato plants by suppressing plant defense responses [55]. However, the sensitivity to saponins might be correlated with the type of sterols present in the membranes of the potential pathogen. The oomycetes have been shown to be insensitive to saponins, probably because their membranes lack 3β-hydroxy sterols [55,57], similar to plant cell membranes. In *P. infestans* there is evidence of genes encoding certain glycoside hydrolases with potential activity against glycoalkaloids, but there is no evidence that deglycosylation takes place [52]. Even though the glycoalkaloids were annotated, the LC-MS data does not allow for determining whether the *P. infestans* infection of the tomato plants produces a significant effect on these metabolites.

On the other hand, the discriminant features M293T15 (*m*/*z* 293.1774) and M221T16 (*m*/*z* 221.1561) are attributed to phytuberin and rishitin, respectively (Table 1). These sesquiterpenoids are antimicrobial phytoalexins produced by plants in response to biotic and abiotic stress [58,59]. Phytuberin and rishitin are reported to be present during the infection of potato plants with *P. infestans* [60,61,62]. Furthermore, the feature M267T14 (*m*/*z* 267.1617) (Table 1) was also annotated as the toxic furanosesquiterpene dihydro-7-hydroxymyoporone, which has been isolated from sweet potato (*Ipomoea batatas*) infected with *Ceratocystis fimbriata* [63].

Some of the significant features annotated as flavonoids are apigenin 7-[rhamnosyl-(1->2)-galacturonide] (M610T24; *m*/*z* 610.1758; [M + NH_3_]^+^) and peonidin 3-(4-sinapoylgentiobioside) (M832T22; *m*/*z* 832.2387; [M + H]^+^). Additionally, naringenin-hexose I (M433T13; *m*/*z* 433.1115) and the catechin 7,4′-dimethyl ether (M317T13; *m*/*z* 317.1020) were annotated (Table 1). The defensive role of flavonoids is less known; however, they are thought to be beneficial to the plant itself as physiologically active compounds, principally to stress protecting agents and play a significant role in plant resistance [4,64,65]. Some metabolomics studies of tomatoes infected with *Botrytis cinerea* showed higher concentrations of flavonoids, such as rutin and quercetin-3-galactoside [37]. Similarly, rutin, saponarin and several kaempferol and related compounds were identified in potato leaves resistant to late blight [66]. Additionally, NMR-based metabolic profiling showed that rutin was the flavonoid that was most expressed in response to *Pseudomonas syringae* infection of tomato [67], and higher levels of rutin are associated with late blight resistance of different cultivars of potato plants [68]. Regarding the specific interaction tomato-*P. infestans*, tomato resistance to phytopathogen was associated with genes involved with reactive oxygen species (ROS) scavenging systems [69]. Besides, the overexpression of the SpWRKY1 gene in tomato regulates antioxidants as flavonoids to reduce ROS accumulation and alleviate cell membrane injury after *P. infestans* infection [70,71].

The other 67 metabolites that were annotated as organic acids, macrolides, alkaloids, and gibberellins are summarized in the Appendix A. The annotation data includes retention time, molecular formulas, exact masses, ion products and metabolite identification levels according to the MSI guidelines [72,73].

### 2.2. MALDI-MS Protocol for Analysis of Infected Tomato

Similar to the LC-MS analysis, we aimed at differentiating the early and late asymptomatic stages and the infection times of the tomato plants infected by *P. infestans* through MALDI-MS metabolic profiles. MALDI-MS is a fast and direct analysis that requires no chromatographic separation [21,75,76]. To optimize the analysis protocol, seven different MALDI matrices were tested. DHB, 9-AA and *trans*-ferulic acid were tested in the positive ion mode, and DMAN, DHB, 4-NA, MBT and ATT were tested in the negative ion mode. The DHB and MBT matrices provided the best MALDI spectra, according to the number and abundance of the ions, over the mass range (600–1500 Da) where there was no matrix effect. Using the selected matrices, characteristic MALDI-MS profiles were obtained from the control, early asymptomatic infection (12 hpi) and late asymptomatic infection stages (96 hpi). Appendix A shows that *m*/*z* 871.4, 909.3, 1034.2 and 1072.1 were the major ions in the positive ion mode. Appendix A shows *m*/*z* 629.8, 661.8, 709.4, 793.4, 815.4 and 837.4 as major ions in the negative ion mode. Due to quite similar MALDI profiles, multivariate analysis was applied in an attempt to differentiate the infected plants into three clusters, similar to what was observed with the LC-MS analysis of the control, early asymptomatic infection (cluster I, 4–36 hpi) and late asymptomatic infection stage (cluster II, 48–96 hpi). 

### 2.3. Multivariate Data Analysis of MALDI-MS

As shown in Figure 4a,b and Figure 5a,b, the MALDI-MS metabolic profiles of early and late asymptomatic infected plants (clusters I and II) and control plants could be differentiated using the PLS-DA analysis. Therefore, such fast MALDI-MS analysis could be used to identify late blight in tomato plants in the absence of symptoms on the basis of their corresponding metabolic profiles. The VIP score plots (Figure 4c,d) show discriminating features for the control, early and late asymptomatic infection clusters that were detected in the positive ion mode (VIP scores > 1). The discriminating ion of *m*/*z* 675.4 decreases throughout the infection, but the discriminating ions of *m*/*z* 1034.2 and 1072.1 decrease in the early asymptomatic infection (Appendix A) and then increase in the late asymptomatic infection stage (see *m*/*z* 1034.2; 1056.2 and 1072.1 ion, Appendix A). These ions are attributed to the protonated molecule and adducts of the glycoalkaloid α-tomatine, that is, [M + H]^+^ of *m*/*z* 1034.2; [M + Na]^+^ of *m*/*z* 1056.2; and [M + K]^+^ of *m*/*z* 1072.1, which is also detected in LC-MS as the ion at *m*/*z* 1078.5436 [M + FA − H]^−^.

The decrease in the relative abundance of α-tomatine adducts in early asymptomatic infection in relation to the control should therefore be related to the successful infection of asymptomatic tomato plants within the first hours. However, there was a subsequent increase of α-tomatine adduct intensities at the late infection stage compared to control, which may be associated with the posterior glycosylation of tomatidine to α-tomatine. This glycosylation appears to be crucial in protecting the cell from the toxic effect of steroidal alkaloids, such as tomatidine, which causes marked developmental defects, including the growth retardation of tomato plants [51].

The most downregulated metabolite in the late asymptomatic infection stage (Appendix A) was detected as the ion of *m*/*z* 871.5 [M + H]^+^ (VIP > 2.5), attributed to pheophytin α. The ions of *m*/*z* 893.3 and *m*/*z* 909.4 correspond to the adducts [M + Na]^+^ and [M + K]^+^, respectively, of pheophytin α (Appendix A). This same ion at *m*/*z* 871.5609 was detected by LC-MS, and its fragmentation pattern allowed us to attribute it as pheophytin α. Pheophytin α is a chlorophyll derivative involved in the electron transfer pathway of photosystem II [77,78]. It has been found that genes involved in photosynthesis and chlorophyll biosynthesis are downregulated upon challenge by virulent and avirulent pathogens [79,80,81], along with the upregulation of defense-related pathways [82].

Likewise, a few metabolites detected in the negative ion mode (score > 0.5) have similar behavior throughout the infection. The intensities of the ions of *m*/*z* 837.4; 839.4 and 831.4 increased in the early stage in relation to control, but those of the ions of *m*/*z* 709.3 and 725.3 are most abundant in the control in relation to the early and late asymptomatic infection stages (Figure 5c,d). The upregulated ions of *m*/*z* 793.4 ([M − H]^−^), *m*/*z* 815.4 ([M – Na − 2H]^−^) and *m*/*z* 837.4 ([M + FA − H]^−^) (Appendix A and Figure 5c) are attributed to the sulfolipid known as 1,2-di-*O*-palmitoyl-3-*O*-(6-sulfoquinovopyranosyl)glycerol (*m*/*z* 793.5109) isolated from *Byrsonima crassifolia* [83]. Sulfoquinovosyl diacylglycerols (SQDGs) and phosphatidylglycerols (PGs) are major classes of the thylakoid membrane lipids in plants [84,85,86]. The increase in SDQGs by *P. infenstans* inoculation might be indicating compositional and/or structural changes of the chloroplast membrane, as reported in tobacco plants inoculated with *Phytophthora parasitica* [87]. 

The score of the VIP plots (Figure 4c,d and Figure 5c,d ) shows that the discrimination power of metabolites is lower in MALDI-MS than in LC-MS, revealing that LC-MS metabolomics provides more significant potential biomarkers. It is likely that MALDI-MS suffers from extensive ion suppression [76,88] due to a lack of chromatographic separation and to interferences of the low molecular weight (<500 Da) ions of the matrix [75]. The MALDI-MS approach could be relevant to future applications for detecting late blight directly on the leaf material through imaging analysis. Although the irregularity of the leaf surface is a limitation for some MSI techniques [89,90], there are some strategies, such as imprinting with several adsorbent materials, that allow the selective adsorption of specific metabolites and that would enable the rapid detection of infection in tomato plants. 

## 3. Materials and Methods

### 3.1. Chemicals

The LC-MS-grade solvents used were acetonitrile, isopropanol (Sigma-Aldrich, St. Louis, MO, USA) was used as an additive for the mobile phase and MALDI matrix solution preparation. The ultrapure water was purified by a Direct-Q water system (Millipore, Bedford, MA, USA). The MALDI matrices were 2,5-dihydroxybenzoic acid (DHB), 2-mercaptobenzothiazole (MBT), 6-aza-2-thiothymine (ATT), 4-nitroaniline (4-NA), N,N,N′,N′-tetramethyl-1,8-naphthalenediamine (DMAN), 9-aminoacridine (9-AA) and *trans*-ferulic acid (Sigma-Aldrich). The lipid standards used for TOF mass calibration were sphingomyelin, 1,2-dipalmitoyl-*sn*-glycero-3-phosphocholine, 2-oleoyl-1-palmitoyl-glycero-3-phosphocholine, and L-α-phosphatidylcholine (Sigma-Aldrich). 

### 3.2. Tomato Plant Samples

The seeds of the Santa Cruz Kada tomato cultivar were obtained from a local market. The seeds were germinated in a germination chamber (120 mm Petri dishes containing wet germination paper) maintained at 18 °C with a 16 h photoperiod for three days. The germinated seeds were planted in 16 cm diameter pots containing a 1:1 mixture of soil and vermiculite and subsequently subirrigated once per day. The 45 plants were maintained at 18 °C, with a 16 h photoperiod, and approximately 60–70% relative humidity (RH).

### 3.3. Pathogen Strain

A culture of *P. infestans*, named as IBSP-34, was obtained from the microorganism collection of the Biological Institute of Sao Paulo, SP, Brazil. The isolate was subcultured on V8 media [91] in 90 mm Petri dishes at 18 °C. After 2–3 weeks, a sporangial suspension was prepared by scraping the surfaces of the colonies with a sterile scalpel, and the mycelia were suspended in sterilized water to produce the infecting solution. The concentration of sporangia in the suspension was adjusted to 1.0 × 10^5^ sporangia mL^−1^ using a Neubauer chamber.

### 3.4. Infection of Tomato Plants with Phytophthora Infestans

After 5–6 weeks, 40 plants were inoculated at the same time with 10 μL of the sporangial suspension of *P. infestans* at four different sites, two on each side of the midrib of the leaf. The infection was carried out by carefully depositing a drop of the sporangial suspension on the leaf with the aid of a micropipette, without mechanically wounding the leaf. The 5 control plants were inoculated with ultrapure water using the same procedure. The experimental design for the infection consisted of two sets of plants that were the initial control (CN) and the inoculated (IN) plants. At each time point after inoculation (4, 12, 24, 36, 48, 60, 72 and 96 hpi), five infected plants were randomly collected and separately analyzed as replicates. 

### 3.5. Sample Preparation 

All the tomato leaves were excised with sterilized scissors and were immediately macerated in liquid nitrogen. A 100 mg aliquot of the crushed powder was vortexed (Multi Reax, Heidolph, Schwabach, Germany), extracted with 1 mL of methanol for 10 min at room temperature, and then centrifuged for 5 min at 12,000× *g* at 20 °C (Centrifuge 5418, Eppendorf, Hamburg, Germany). The supernatants were stored at −20 °C until analysis. For LC-MS analysis, 50 µL of the supernatant was diluted with 950 µL of methanol in a vial. For MALDI-MS analysis, 1 µL of the supernatant was deposited on the MALDI plate.

### 3.6. Untargeted Analysis of Metabolites

#### 3.6.1. UHPLC-Q-TOF-MS Analysis 

The extracts of infected and noninfected leaves were analyzed with a 1290 Infinity UHPLC coupled to an Agilent 6550 iFunnel Q-TOF LC-MS system with Agilent Dual Jet Stream electrospray ionization technology (ESI, Agilent Technologies, Santa Clara, CA, USA). For metabolite separation, a Kinetex XB-C18 Core-Shell column (2.1 × 150 mm, 1.7 μm, 100 Å; Phenomenex Inc., Torrance, CA, USA) was used and maintained at 40 °C. The mobile phases were A (0.1% aqueous formic acid) and B (0.1% formic acid in methanol). The chromatographic gradient of B was increased from 5% to 95% over 18 min and maintained for 7 min at 95% with a flow rate of 0.35 mL·min^−1^. Subsequently, the initial conditions were reached in 8 min, and the column was equilibrated for 7 min. The injection volume was 2 µL. To avoid degradation, the samples were maintained at −20 °C in a freezer, and prior to analysis they were placed in an autosampler maintained at a room temperature of approximately 21 °C. Prior to injection, the needle was washed for 20 s with a mixture of H_2_O:ACN:IPA (4:4:2) using the flush port. The external needle wash volume was 150 µL.

The ESI source was used in the positive or negative ion mode in the following conditions: drying gas temperature 250 °C, drying gas flow rate 14.0 L·min^−1^, sheath gas temperature 250 °C, sheath gas flow rate 10.0 L·min^−1^, nebulizer gas 45 psig, and capillary voltage +3.5 kV and −3.5 kV for the positive and negative ionization mode, respectively. The Q-TOF parameters were acquisition rate of 1.0 spectra/s over the *m*/*z* 100−1700 amu range, skimmer voltage 65 V, octopole RF 750 V, fragmentor 150 V and nozzle voltage 350 V. The multichannel plate detector voltage was 650 V, and the photomultiplier tube voltage was 700 V. The pulser was set to a pulse width of 125 counts/pulse, with a pulse width of 25 counts. A second reference sprayer orthogonal to the sample sprayer in the electrospray source was used to introduce a reference solution for accurate mass determinations. The reference mass ions were of *m*/*z* 121.0509 and *m*/*z* 922.0098 in ESI (+) and *m*/*z* 119.0363 and *m*/*z* 966.0007 in ESI (−). The detection window for the reference masses was set to 10 ppm, with an average of 10 scans and a minimum peak height of 100 counts/s. A mass calibration was performed with an Agilent tune mix from 100 to 1600 Da. The data were acquired in profile mode using high resolution mode (2 GHz).

First, an experiment in full scan mode over the entire mass range was performed, and then two fragmentation experiments were carried out. The auto MS/MS experiments were performed with fixed energies of 30, 40 or 50 eV, in which a first step was to select the 10 most intense ions per scan to produce the MS^2^ spectra. The auto MS/MS experiments that were performed in the variable energy mode used scan speeds that varied based on the precursor abundance and used energies based on the precursor mass-to-charge. The collision energies applied were 6.5 V/100 Da and were offset by 2.0 V. The QC samples consisted of a pool of all the different inoculation times and controls and were analyzed at the beginning and at the end of each batch and after every 10 injections.

#### 3.6.2. MALDI-MS Profile Analysis

The extracts (1 μL) were directly smeared onto a 384-position MALDI target plate (Bruker Daltonics, Bremen, Germany). After drying, the spots were immediately overlaid with 1 μL of the matrix solution: 2,5-dihydroxybenzoic acid (DHB) for the positive ion mode and 2-mercaptobenzothiazole (MBT) for the negative ion mode. All MALDI matrices used for method optimization (DHB, ATT, 4-NA, MBT, 9AA and *trans*-ferulic acid) were prepared at a concentration of 10 mg·mL^−1^ in methanol/water (80:20 *v*/*v*). The DMAN matrix solution only was prepared at a concentration of 10 mg·mL^−1^ in methanol/water (90:10 *v*/*v*) because of its low solubility. Tuning of the smart beam laser position and energies, as well as electronic adjustment of the lens, was performed by Bruker when necessary. The external TOF calibration was made using 1 μL of a lipids mixture composed of sphingomyelin, 1,2-dipalmitoyl-*sn*-glycero-3-phosphocholine, 2-oleoyl-1-palmitoyl-glycero-3-phosphocholine and l-α-phosphatidylcholine, followed by 1 μL of DHB applied in the same conditions as the samples [92]. 

The MALDI-MS analyses were performed on a Bruker Autoflex III MALDI–TOF/TOF mass spectrometer equipped with a 334 nm smart beam laser. The metabolite profiles were acquired in the TOF reflector mode, using the positive or negative ion mode. The accelerating voltage was +20 kV for positive and −20 kV for the negative ion mode, with delayed extraction of 260 ns for both. Each spectrum was manually collected as an average of 5000 laser shots (1000 laser shots at five different positions on the same spot). The laser energy was set at 70% for spectra acquisition. The *m*/*z* 600–1500 range was used for metabolite profile acquisition of both positive and negative ion modes. The spectra were acquired in triplicates via the AutoExecute tool of the Flexcontrol acquisition software (version 2.4; Bruker-Daltonik GmbH, Bremen, Germany). 

### 3.7. Multivariate Data Analysis

#### 3.7.1. LC-MS-Based Metabolomics Data

The LC-MS data processing was performed using MassHunter Qualitative Analysis (Agilent Software B07.00, Santa Clara, CA, USA), in which raw original data was converted to the *mzData* format. The *mzData* files were uploaded to XCMS online for further data processing (https://xcmsonline.scripps.edu/). The XCMS software was used for feature detection, retention time correction, feature alignment and univariate statistical analysis [93]. The data were analyzed as pairwise jobs, with the following settings: centWave feature detection with 5 ppm of maximal tolerated *m*/*z* deviation; minimum peak width 5 s; maximum peak width 20 s; signal/noise threshold 6; noise filter abundance 0; prefilter abundance 100; mzdiff 0.01, and integration method type 1. Obiwarp retention time correction with 1 *m*/*z* step size (profStep) was used to generate the profiles. Other parameters were alignment: mzwid 0.015; minfrac 0.5; mz width: 0.015; bw: 5. An unpaired parametric t test (Welch test) was performed to identify significant features with a p value threshold of 0.05 and a fold change threshold (highly significant features) of 1.5 [94]. The *.csv* file from the XCMS processing was uploaded to MetaboAnalyst 3.0 (http://www.metaboanalyst.ca/) for multivariate statistical data analysis. The file comprised a list of features (*m*/*z*, retention times and intensities) for all samples from infected and noninfected leaves. The data processing applied an integrity check, missing value check, data filter, and normalization before statistical analysis [95,96]. The presence of missing values or features with constant values (i.e., all zeros) was checked, and data filtering using the interquantile range (IQR) was applied to remove variables close to the baseline or detection limits and variables with near-constant value [97]. 

#### 3.7.2. MALDI-MS Profile Data

The analysis of MALDI data was conducted in three distinct steps: (1) preprocessing, (2) processing and (3) statistical analysis. The raw spectra were preprocessed in the FlexAnalysis software (Bruker-Daltonik GmbH, Bremen, Germany) after baseline subtraction for background removal, alignment of the spectra scale, ion selection with an S/N ratio greater than 3 and normalization of intensities. Data processing was performed before multivariate analysis for the metabolite profiles in MetaboAnalyst 3.0. The uploaded files (.*csv* format) comprised a list of features (*m*/*z* and relative intensities). The ions were realigned within a tolerance of *m*/*z* 0.4 (0.4 Da) to remove ions that appeared in less than half of the samples in each group [92]. The presence of missing values or features with constant values (i.e., all zeros) was checked, and data filtering using relative standard deviation (RSD) was applied to remove variables close to baseline or detection limit and variables with near-constant values. 

Both LC-MS and MALDI-MS data were normalized by sum for adjustment of the differences among samples and Pareto scaling (mean-centered and divided by the square root of the standard deviation of each variable) was used to make individual features more comparable [92]. To discriminate the infection times of the tomato plants with *P. infestans* based on their LC-MS and MALDI-MS metabolite profiles, principal component analysis (PCA), partial least squares discriminant analysis (PLS-DA) and orthogonal partial least squares discriminant analysis (OPLS-DA) was performed on the data using MetaboAnalyst 3.0. To identify the molecular features related to variation between groups of samples, the corresponding loading plots and the variable importance in projection (VIP) were applied [98]. The validation of the classification models obtained by multivariate analysis was made to confirm the capability of classification and prediction of the models. The data were permuted 100 times, and Q2 and R2Y were used as quality-of-fit criterion [99]. 

### 3.8. Annotation of Metabolites 

The major discriminant features of the LC-MS analyses were selected out of the multivariate analysis, and their annotation was made according to the exact mass (*m*/*z*) of the protonated or deprotonated ion and their fragmentation spectra. The *.txt* data archives with *m*/*z* and relative abundance of the ions obtained from MS and MS/MS spectra were uploaded to MS-FINDER software ver. 2.26 (http://prime.psc.riken.jp/Metabolomics_Software/MS-FINDER/), which provides molecular formulas of the precursor ions based on accurate mass, isotope ratio, and product ion spectra [100].

The Metlin (http://metlin.scripps.edu), ChemSpider (www.chemspider.com), MassBank (http://www.massbank.jp/), HMDB (http://www.hmdb.ca/), and LipidMaps (http://www.lipidmaps.org/) spectra databases and a comparison with fragmentation profiles of previously reported metabolites from tomato infection were also used for confirmation of the metabolite annotation [1,38,51,101,102,103]. 

## 4. Conclusions

The LC-MS metabolites profiles discriminate between early and late asymptomatic infection, and between each infection time in the infected tomato plants and identified major metabolites that are altered in late blight. Metabolites detected via LC-MS operating in the negative ion mode provided more discriminant clusters compared to those detected in the positive ion mode. The annotated metabolites correspond to tricarboxylic acid (TCA) cycle secondary metabolites and include terpenoids, flavonoids, alkaloids, saponins, sesquiterpenes and glycoalkaloids. We found the metabolite tomatidine to be an important biomarker of infection because it is produced by the action of the fungal pathogen enzymes. Also, we found that saponins might be early infection metabolite markers because their abundance increases between 4 and 36 hpi as specific response to the type of sterols present in the pathogen membrane. We found the metabolite isocoumarin (M301T17) as a good infection marker because its abundance increases linearly along the post infection time. These metabolites could be relevant in future applications to detect late blight directly on the asymptomatic leaf material through imaging analysis using DESI-MS or MALDI imaging.

Additionally, the metabolite profiles obtained by MALDI (±)-MS, associated with multivariate analysis, have provided late blight detection of tomato plants in early and late asymptomatic infection. The major discriminant metabolites are α-tomatine, pheophytin α and 1,2-di-*O*-palmitoyl-3-*O*-(6-sulfoquinovopyranosyl)glycerol, but α-tomatine has an important role in infection control because it decreases within the first hours and increases in the late asymptomatic infection stage. MALDI (±)-MS seems to offer a rapid and effective method to detect late blight in asymptomatic tomato plants and therefore it could function as a suitable guide for the management of sanitary defense approaches.

## Figures and Tables

**Figure 1 molecules-23-03330-f001:**
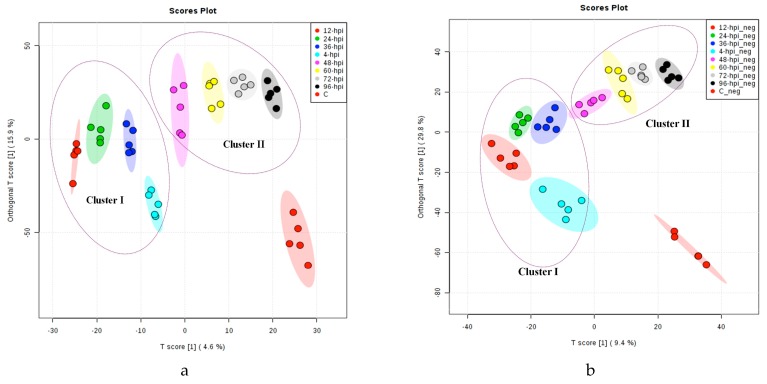
OPLS-DA score plots for the eight post-infection time points (4, 12, 24, 36, 48, 60, 72 and 96 hpi) and the initial control: (**a**) LC-ESI (+)-MS and (**b**) LC-ESI (−)-MS. VIP score-plot derivate of PLS-DA analysis (**c**) LC-ESI (+)-MS and (**d**) LC-ESI (−)-MS. The coding M and T after the number indicates nominal mass and retention times, respectively, of each feature. Cluster I includes the infection times of 4–36 hpi. Cluster II includes the infection times of 48–94 hpi. Coding 4-hpi indicates positive ionization mode analysis for the samples collected at four hours post infection. Coding 4-hpi_neg means negative ionization mode analysis of the samples collected at four hours post infection. Coding C and C_neg means the analysis of the controls in positive and negative ionization modes, respectively. Variable Importance in Projection (VIP), is a weighted sum of squares of the PLS loadings taking into account the amount of explained Y-variation, in each dimension. VIP scores are calculated for each component and were used average of two components to calculate the feature importance. The color scale depends on the range intensity of the metabolites in all samples and the media intensity of the samples the same time.

**Figure 2 molecules-23-03330-f002:**
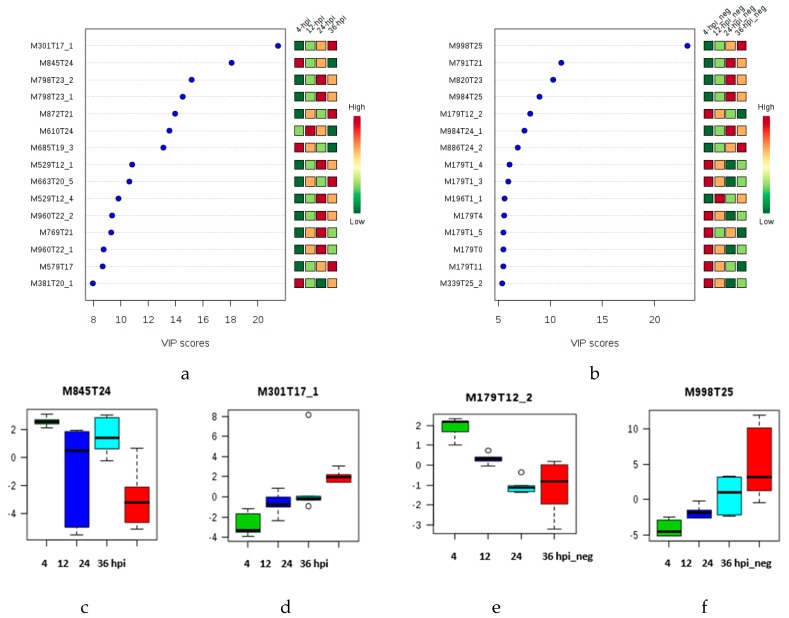
VIP score-plots derived from the PLS-DA analysis, displaying the discriminant features at the early asymptomatic infection stage (cluster I: 4, 12, 24 and 36 hpi): (**a**) LC-ESI (+)-MS and (**b**) LC-ESI (−)-MS. Coding of M and T after the number respectively indicates nominal mass and retention time of each feature. (**c**–**f**). Box plots from s-plot of the molecular features M845T24 (*m*/*z* 845.4816), M301T17 (*m*/*z* 301.1379), M179T12 (*m*/*z* 178.9792) and M998T24 (*m*/*z* 997.5185), respectively. In the box plot, Y axis represents intensity of the metabolites as quartile for each sample group related to all data set. The range of the vertical scale is from the minimum to the maximum value of the selected group, or, to the highest or lowest of the displayed reference points, median, and 95% confidence interval of the mean. Coding 4-hpi means positive ionization mode analysis of the samples collected at four hours post infection. Coding 4-hpi_neg means negative ionization mode analysis of the samples collected at four hours post infection. Coding C and C_neg indicates the analysis of the controls in positive and negative ionization modes, respectively.

**Figure 3 molecules-23-03330-f003:**
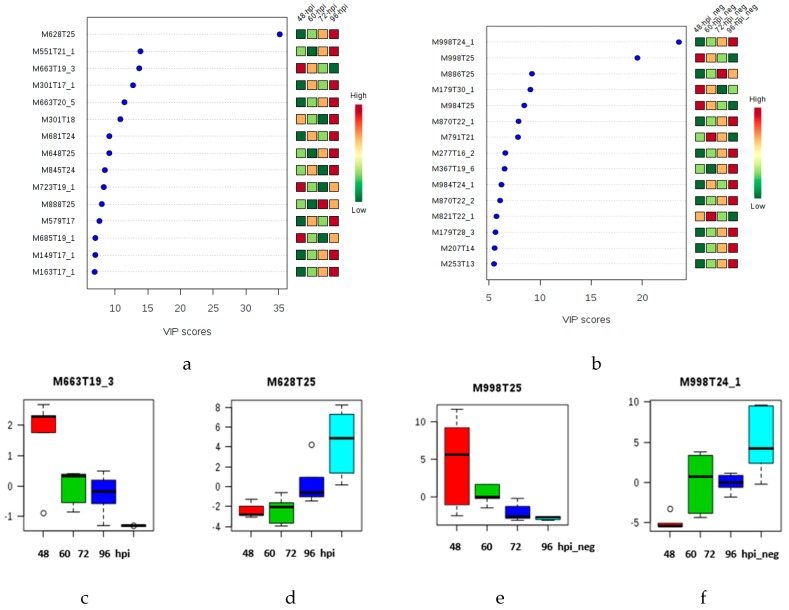
VIP score-plots derived from the PLS-DA analysis displaying discriminant features at the late infection stage (cluster II: 48, 60, 72 and 96 hpi): (**a**) LC-ESI (+)-MS and (**b**) LC-ESI (−)-MS. Coding of M and T after the number respectively indicates the nominal mass and retention time of each feature. (**c**–**f**). Box plots from s-plot of the molecular features M663T19 (*m*/*z* 663.4508), M628T25 (*m*/*z* 628.3657), M998T24 (*m*/*z* 997.5185) and M998T25 (*m*/*z* 997.5184), respectively. In the box plot, Y axis represents intensity of the metabolites as quartile for each sample group related to all data set. The range of the vertical scale is from the minimum to the maximum value of the selected group, or, to the highest or lowest of the displayed reference points, median, and 95% confidence interval of the mean. Coding 48-hpi means positive ionization mode analysis of the samples collected at forty-eight hours post infection. Coding 48-hpi_neg means negative ionization mode analysis of the samples collected at forty-eight hours post infection. Coding C and C_neg means the analysis of the controls in positive and negative ionization modes, respectively.

**Figure 4 molecules-23-03330-f004:**
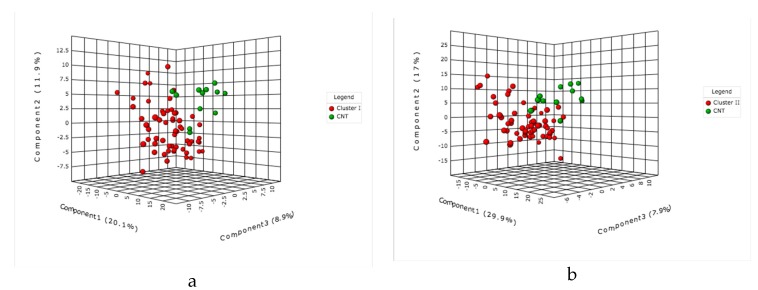
Multivariate partial least squares discriminant analysis applied to the MALDI (+)-MS data: (**a**) Early asymptomatic infection and control and (**b**) Late asymptomatic infection and control. The VIP plots (**c**) Early asymptomatic infection and control and (**d**) Late asymptomatic infection and control.

**Figure 5 molecules-23-03330-f005:**
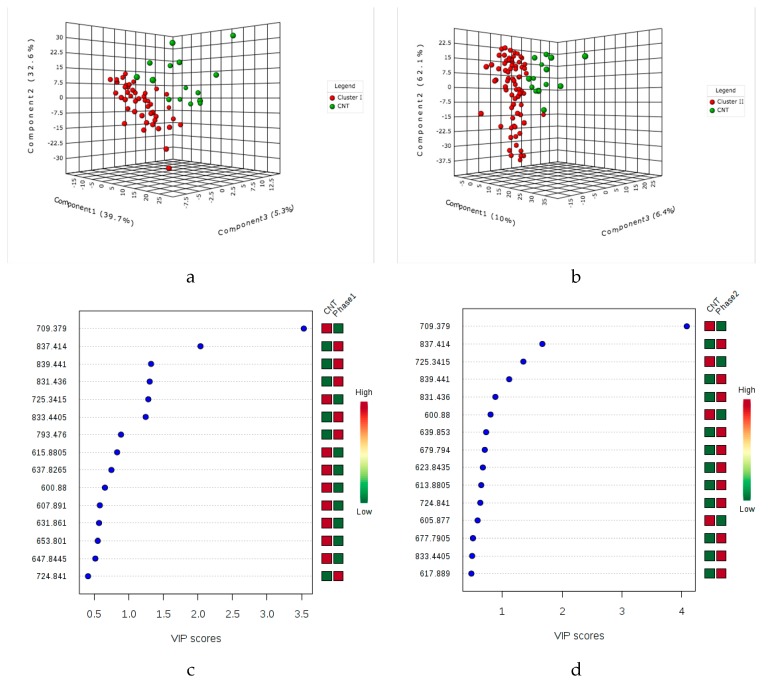
Multivariate partial least squares discriminant analysis applied to the MALDI (−)-MS data: (**a**) Early asymptomatic infection and control and (**b**) Late asymptomatic infection and control. VIP plots (**c**) Early asymptomatic infection and control and (**d**) Late asymptomatic infection and control.

**Table 1 molecules-23-03330-t001:** Principal metabolites associated with tomato late blight annotated with LC-MS/MS data. Annotation level: identified metabolites (level 1), putatively annotated compounds (level 2), putatively characterized compound classes (level 3), and unknown compounds (level 4) [73,74].

Molecular Feature	RT (min)	Metabolite	Molecular Formula	Annotation Level	Theor. (*m*/*z*)	Found (*m*/*z*)	AME (ppm)	Adducts
M609T9	9.06	Quercetin 7-(rhamnosylglucoside)	C_27_H_30_O_16_	2	609.14555	609.1464	1.4	[M − H]^−^
M1095T10	9.85	Hydroxytomatine isomer I	C_50_H_83_NO_22_	3	1094.53832	1094.5382	0.1	[M + FA − H]^−^
M1077T12	11.52	Dehydrotomatine	C_50_H_81_NO_21_	2	1076.52776	1076.5277	0.1	[M + FA − H]^−^
M1109T12	11.58	UGA11	C_52_H_87_NO_24_	3	1108.55397	1108.5541	0.1	[M − H]^−^
M1121T12_1	11.72	UGA28	C_52_H_83_NO_25_	3	1120.51759	1120.5120	5.0	[M − H]^−^
M1079T12	11.94	Tomatine	C_50_H_83_NO_21_	2	1078.5440	1078.5436	0.4	[M + FA − H]^−^
M433T13_2	12.86	Naringenin-hexose I	C_21_H_22_O_10_	3	433.11347	433.1115	4.5	[M − H]^−^
M317T13_2	13.09	Catechin 7,4′-dimethyl ether	C_17_H_18_O_6_	2	317.10251	317.1020	1.6	[M − H]^−^
M293T15_1	14.51	Phytuberin	C_17_H_26_O_4_	2	293.17528	293.1774	7.2	[M − H]^−^
M221T16_2	16.30	Rishitin	C_14_H_22_O_2_	2	221.15415	221.1561	8.8	[M − H]^−^
M737T20	19.55	Tuberoside J	C_39_H_64_O_14_	2	737.41121	737.4186	10.0	[M − H]^−^
M414T20	19.59	Tomatidine	C_27_H_45_NO_2_	2	414.3372	414.3385	3.1	[M − H]^−^
M791T21	21.46	Triterpenoid saponins- Sapimukoside J	C_44_H_72_O_12_	2	791.49455	791.4955	1.2	[M − H]^−^
M832T22_2	22.21	Peonidin 3-(4-sinapoylgentiobioside)	C_39_H_43_O_20_	2	832.24259	832.2387	4.7	[M + H]^+^
M794T22_1	22.24	1,2-Di-O-palmitoyl-3-O-(6-sulfoquinovopyranosyl)glycerol	C_41_H_78_O_12_S	2	793.51357	793.5109	3.4	[M − H]^−^
M798T23_2	22.58	Triterpene saponins- UNPD101109	C_43_H_72_O_13_	2	797.50511	797.5049	0.3	[M + H]^+^
M820T23	22.69	Sulfoquinovosyldiacylglycerols	C_43_H_80_O_12_S	3	819.5289	819.5269	2.4	[M − H]^−^
M743T23_3	22.83	3-*O*-a-l-Arabinopyranosylproanthocyanidin A5	C_35_H_32_O_16_	2	743.13788	743.1395	2.2	[M + Cl]^−^
M610T24	24.07	Apigenin 7-[rhamnosyl-(1->2)-galacturonide]	C_27_H_28_O_15_	2	610.17719	610.1758	2.3	[M + NH_4_]^+^
M845T24	24.40	Triterpene saponins- Cyclopassifloside III	C_43_H_72_O_16_	2	845.48986	845.4816	9.8	[M + H]^+^
M984T24_1	24.49	Triterpene saponins	C_48_H_84_O_18_	3	983.53461	983.5393	4.8	[M + Cl]^−^
M984T25	24.87	Triterpene saponins	C_48_H_84_O_18_	3	983.53461	983.5392	4.7	[M + Cl]^−^

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
