# Peer review of "Metabolomics of *Solanum lycopersicum* Infected with *Phytophthora infestans* Leads to Early Detection of Late Blight in Asymptomatic Plants"

_molecules, 2018, doi:10.3390/molecules23123330_

Round 1

Reviewer 1 Report

p { margin-bottom: 0.1in; line-height: 115%; background: transparent none repeat scroll 0% 0%; }a:link { color: rgb(0, 0, 128); text-decoration: underline; }a:visited { color: rgb(128, 0, 0); text-decoration: underline; }

Dear Authors,

I think this is a well-designed, excellent piece of work. Consider the following minor corrections that are necessary before acceptance for publication.

Major issues.

When determining the significance for each test, divide your p treshold with the number of performed statistical tests (also known as Bonferroni correction) to maintain the overall chance of false positive detection of phenomena at an acceptable level.

What is known about the P. infestans tolerance of your variety? Do we see a the metabolome change of a possibly succesful defense reaction, or your plants will develop disease symptoms with time? The infection is symptomless throughout the examination period. When would symptoms usually appear? Add some additional data.

Minor issues.

Why are QC sample PCA score values not centered around zero (Fig. S3)? If QC samples are the average of all samples, this should be the case in all PCs.

L95-96: regarding Fig S3-S4: Instead of clusters, I’d rather say that the time gradient is clearly visible in the PCA score plot. Only a minor proportion of your overall variance is covered in the score plot (which is expectable for such a complex matrix). Are there other spectacular projections? Consider adding them to supplementary material.

A careful look at Fig S5 provides rather three stages: 4-12, 24-36, and 48-96. In other words, a whole cluster (in the middle of the heatmap) of metabolites increase in concentartion in the 24-36 sample, compared to the 4-12 hpi ones. What supports division into two sets? To me, division into two sets seems somewhat arbitrary.

FigS5: color code of “C” and “12-hpi” can not be distinguished, please update the figure.

Change the order of the x axis in the small heatmaps in Fig.1 c-d to C, 4, 12, etc. It is disturbing that 4 is between 36 and 48.

Table 1: What do the annotation levels (2, 3) mean? Correct to [M+Cl]-.

Table S1: I see some unlikely annotations: Notoginsenoside N etc. Were all these metabolites described from tomato by other research? You are not using authentic standards, these annotations are putative identifications. In these cases you should rather write “unidentified triterpene x-hexoside” or something similar, unless you provide additional data on identity. What is “ketals” or “macrolides”?

Fig 3 is broken: there is a subplot at page 5 bottom right.

Best regards.

Author Response

 Report (Reviewer 1)

Major issues.

Point 1. When determining the significance for each test, divide your p treshold with the number of performed statistical tests (also known as Bonferroni correction) to maintain the overall chance of false positive detection of phenomena at an acceptable level.

Response 1: We did check for the chance of false positive by Bonferrori correction and found an acceptable level. The text “Bonferroni correction was used to adjust p-value to (0.01/45 = 0.0002)” was added to the figure legend S1 and S2.

Point 2a. What is known about the P. infestans tolerance of your variety?

Response 2a. According to Zuluaga and Schoina, the interval which comprises the hemibiotrophic life cycle of P. infestans (first biotrophic and then necrotrophic) in tomato is of about 96 hours post inoculation (hpi). However, marked differences in the duration of both biotrophic and necrotrophic growth, depending on the host-P. infestans isolate interaction [1,2].

Although, early and late stages of the infection were detected on the basis of the metabolomic profile, no necrosis was observed in the tissue, up to the last infection time evaluated (96 h). This lack of symptoms may be due to diminished aggressiveness of the P. infestans strain towards tomato as it was isolated from potato plants. Therefore, symptoms may appear later. Also, the Santa Cruz Kada cultivar is partially resistant to P. infestans [3]. Thus, partially resistant cultivar infected with P. infestans strains from potato can provide a good model to study asymptomatic interaction.

Point 2b. Do we see a metabolome change of a possibly successful defense reaction, or your plants will develop disease symptoms with time?

Response 2b. Our results showed that tomato plants of Santa Cruz Kada variety infected with P. infestans were macroscopically asymptomatic during the first 96 hours after inoculation. Considering that the symptoms can appear after 120 to 144 hpi and P. infestans strain isolated from potato can be less aggressive, we still detected a metabolomic change related to disease symptoms, but very early. We can see the progression of the metabolites related to disease along the post infection time. However, we don’t have information of this specific interaction after 96 hpi.

Point 2c) The infection is symptomless throughout the examination period. When would symptoms usually appear? Add some additional data.

Response 2c. As it was mentioned by Schoina, the duration of both biotrophic and necrotrophic steps in the life cycle of P. infestans, depending on the host-P. infestans isolate interaction [2]. For example, studies performed by Åsman on potato, show that the biotrophic stage occurs one day post inoculation (dpi) with P. infestans, but 2 dpi is believed to be the transition point from biotrophy to necrotrophy, and 3 dpi is considered the necrotrophic stage [4], that is, at 72 h symptoms of infection should be observed in potato plants.

Our experimental design only contemplated the analysis until 96 hpi which is related to lack of symptoms; however, previous studies of our group (as yet unpublished) show that the first symptoms of the infection on tomato (Santa Cruz cultivar) appear around 120-144 hours after inoculation with P. infestans (UR-24 strain). The Figure 1 shows the first symptoms of the tissue necrosis after 144 hours post inoculation of UR-24 strain of P. infestans on tomato Santa Cruz variety.

Figure 1. Tomato plant infected by P. infestans. Macroscopically symptoms.

Minor issues.

Point 3. Why are QC sample PCA score values not centered around zero (Fig. S3)? If QC samples are the average of all samples, this should be the case in all PCs.

Response 3: We prepared the QC sample by diluting 5 µL of each post infection sampling time and control to 1000 µL with SOLVENT. Therefore, the QCs were less concentrated than the samples, so they do not appear in the center of the PCA. However, we did 10 injections of QCs for each batch and all of the injections were grouped in the same cluster, showing that the QCs did not suffer batch effect.

Point 4. L95-96: regarding Fig S3-S4: Instead of clusters, I’d rather say that the time gradient is clearly visible in the PCA score plot. Only a minor proportion of your overall variance is covered in the score plot (which is expectable for such a complex matrix). Are there other spectacular projections? Consider adding them to supplementary material.

Response 4: The variance of the other spectacular components was much lower and doesn’t shown great differences in the cluster separation which is expectable for a complex matrix. Because of the lower variance covered, due to the complex matrix, we didn’t add it to the supplementary material, but instead we did multivariate analysis with minor groups of samples to reveal more differences between clusters.

Point 5. A careful look at Fig S5 provides rather three stages: 4-12, 24-36, and 48-96. In other words, a whole cluster (in the middle of the heatmap) of metabolites increase in concentartion in the 24-36 sample, compared to the 4-12 hpi ones. What supports division into two sets? To me, division into two sets seems somewhat arbitrary.

Response 5: We used OPLS-DA score plot to define division into two clusters, because OPLS-DA reduce data variance to a few components with major variances. Through the quadrants of the score plot, we can deduce three groups that are initial control, early infection (4 to 36 hpi, cluster I) and late infection (48 to 96 hpi, cluster II). A careful look of the HCA can lead to point out a tendency of a middle cluster related to 24-36 samples, but this group is not completely separated from the 4-12 cluster. HCA uses all variables, therefore offers an additional information about the similarities between samples and clusters. We are able to see similarities between the 4-12 hpi and the 24-36 hpi samples because there is a connection.

Point 6. FigS5: color code of “C” and “12-hpi” can not be distinguished, please update the figure.

Response 6. The color code of “C” was changed in the Figure S5 and S6.

Point 7. Change the order of the x axis in the small heatmaps in Fig.1 c-d to C, 4, 12, etc. It is disturbing that 4 is between 36 and 48.

Response 7. Done. See the the following figure for confirmation

Point 8. Table 1: What do the annotation levels (2, 3) mean? Correct to [M+Cl]-.

Response 8: The annotation level corresponds to the level of identification of the metabolites according to the Metabolomics Standard Initiative (MSI). Currently, four levels of metabolite identifications can be found in the published metabolomics literature. They include [5]:

Identified compounds.

Putatively annotated compounds (e.g. without      chemical reference standards, based upon physicochemical properties and/or      spectral similarity with public/commercial spectral libraries).

Putatively characterized compound classes      (e.g. based upon characteristic physicochemical properties of a chemical      class of compounds, or by spectral similarity to known compounds of a      chemical class).

Unknown compounds—although unidentified or      unclassified these metabolites can still be differentiated and quantified      based upon spectral data.

The text “Annotation level: identified metabolites (level 1), putatively annotated compounds (level 2), putatively characterized compound classes (level 3), and unknown compounds (level 4)” was added to the table legend for clarify [5,6].

The adducts [M+Cl]-  in      the table 1 and S1 were corrected.

Point 9a. Table S1: I see some unlikely annotations: Notoginsenoside N etc. Were all these metabolites described from tomato by other research? You are not using authentic standards; these annotations are putative identifications. In these cases, you should rather write “unidentified triterpene x-hexoside” or something similar, unless you provide additional data on identity.

Response 9a. According to the Minimal Reporting Standards in chemical analysis of Metabolomics Standard Initiative (MSI), the level two (level 2) correspond to putatively annotated compounds. In the case of identified metabolites (level 1), it is mandatorily necessary to employ a standard for the correct identification of each compound. This is the reason why the level of annotation is two in the case of most compounds whose mass spectra (MS and MSMS) were compared in databases (Metlin, HMDB, MS-Finder, LipidsMaps) and yielded information about a certain metabolite. Most of the annotated compounds (level 2) have been reported in tomato plants or in other plant-pathogen interactions.

Point 9b. What is “ketals” or “macrolides”?

Response 9b. In the table S1 ketals, macrolides, triterpene saponins, glycosyldiacylglycerols and other, correspond to the level 3: putatively characterized compound classes. The annotation level is described in the tables 1 and S1.

Point 10. Fig 3 is broken: there is a subplot at page 5 bottom right.

Response 10: Yes, it was corrected.

Reviewer 2 Report

This manuscript describes application of the LC-MS and MALDI-MS to analyze the effects of Phytophthora infestans on Solanum lycopersicum plants at eight time points after inoculation. The multivariate data analysis is employed to analyze the obtained metabolic data. The differential analysis of inoculated vs. control leaves revealed a broad set of MS signals and some of them were identified. The Authors conclude that metabolic profiling might lead to identification of Late blight in tomato.

Comments

In my opinion a clear statement is missing in the section Conclusions – are there any metabolic markers of Late blight disease detected here ? Is their profile different for the sebsequent post-inoculation time points ?

The manuscript is hard to read and difficult to follow – it is because the text is too dense. Especially the section between lines 97 – 173 has to be rewritten. Instead of the extensive list of observables (eg., l.106-110, l. 142– 145 etc.) the Authors should comment the results giving just a selected example; moreover whenever possible it would be informative to see here the (putative) name of the identified compound besides its identification code or m/z value.

Several statistical methods are used in this manuscript. For the general audience the rationale of their application has to be described besides the acronym (e.g., VIP scores, PLS-DA) in the section Results.

Since two MS techniques are used in this manuscript I would appreciate the comparison of the two sets of the obtained data using the Venn diagram.

An application of MALDI MS to analyze effects of Late blight on tomato decribed by Laurindo et al., 2018 should be mentioned in the Introduction. Results described in [7] should be extensively commented in the context of the data obtained in this report. Interestingly, report by Laurindo et al., 2018 is included in the reference list [7] but still it is not discussed in this context.

Moreover, it would be interesting to see a brief discussion of the metabolic data obtained in this manuscript in light of the transcriptomic data reported by Jiang et al., Plant Cell Rep. 2018, Cui J et al., Theor Appl Genet. 2018 and Cui et al., Plant J. 2017.

Formal comment – I would suggest to replace the term ‘feature’ with ‘signal’ or ‘MS signal’

Minor remarks

l.29 – ‘late blight’ or ‘Late blight’ - please unify the notation here and through the entire manuscript

l.30 – why Phytophthora infestans and Late blight are missing in the key words ?

l.106 and below – is the accuracy of the m/z values correct ? were the HR measurement performed here ?

l. 139 – please rewrite the sentence – it is unclear ‘The PLS-DA plots of Figures S8a and S8b verify the closeness between the times of the second cluster.’

l.312 – sphingomyelin is not a phospholipid

l.320 – were plants cultivated in the greenhouse or rather upon the controlled light regime?

l.322 – delete ‘pure’

l.324 – ‘V8 media’ – describe the composition or provide the reference

l.335 – correct the description: were the entire plants used for the metabolic profiling or just selected leaves? which leaves were collected – only the infected ones ? Were any necrotic changes observed ?

l.341 – replace ‘12,000 g’ with ‘12,000 x g

l.341-4 – delete ‘extract’

l.352 – delete ‘mobile phase’

l.346 – was any external standard used for UPLC-MS ?

l.670 – delete ‘61’

Fig. 1 –a brief explanation of how the VIP plots were generated should be included either in the legend or the main body of the text. Do the color scale presents the intensity of the particular signals – explain in the text. Please explain the Control (mock treatment? Time point ?)

Fig. 2 and 3 – the legend should clearly describe the content of panels c-f, what are the units of Y axis ? formatting of these panels requires correction

Fig. 4 – MALDI profiles should be transferred to the supplement

Table 1 – meaning of the ‘Annotation level’ has to explained in the legend. Moreover, comas have to be replaced with dots.

Fig.S1 and S2 – legends are not complete – color code (part A) as well as explanation of the meaning of the red circles (part B) have to be provided

Fig.S3 and S4 – as above –  a brief explanation how the Quality Control was obtained is missing. Control has to be described too, additionally abbreviation for negative control (CN) is not used in the panels. Moreover, notation ‘C_neg’ should be consequently used through the entire manuscript

Table S1 – legend - please correct ‘chamical analysis’

Author Response

 Report (Reviewer 2)

Comments

Point 1. In my opinion a clear statement is missing in the section Conclusions – are there any metabolic markers of Late blight disease detected here ? Is their profile different for the sebsequent post-inoculation time points ?

Response 1. The next text was added. We found the metabolite tomatidine to be an important biomarker of infection because it is produced by the action of the fungal pathogen enzymes. Also, we found that saponins might be early infection metabolite markers because their abundance increases between 4 and36 hpi as specific response to the type of sterols present in the pathogen membrane. We found the metabolite isocoumarin (M301T17) as a good infection marker because its abundance increases linearly along the post infection time (line 503).

Point 2. The manuscript is hard to read and difficult to follow – it is because the text is too dense. Especially the section between lines 97 – 173 has to be rewritten. Instead of the extensive list of observables (eg., l.106-110, l. 142– 145 etc.) the Authors should comment the results giving just a selected example; moreover, whenever possible it would be informative to see here the (putative) name of the identified compound besides its identification code or m/z value.

Response 2. The entire manuscript was submitted to style correction by the American journal experts (see attached certificated). Also, the section between the lines 97-173 was modified. We introduced the name of some annotated metabolites and suppress the extensive list of molecular features and m/z values. See part of the modified text:

In the first step of the statistical processing, the principal component analysis (PCA), a technique used for the dimensionality reduction of multivariate data whilst preserving most of the variance [1], it was applied to LC-MS metabolic profiles data in order to find clusters of samples. The PCA score plot (Figures S3 and S4) clearly shows three clusters of the samples according to the different infection times, corresponding to early asymptomatic infection (4, 12, 24 and 36 hpi), late asymptomatic infection (48, 60, 72 and 96 h) and the initial control.

Secondly, multivariate analysis of OPLS-DA was applied, and it revealed significant differences between the controls and early- and late-infected asymptomatic plants, along with additional information regarding significant molecular features of the infection stages. In the OPLS-DA plots (Figures 1a and 1b), three different clusters in both the positive and negative ion modes corresponding to the initial control, early infection (4 to 36 hpi, cluster I) and late infection (48 to 96 hpi, cluster II) were delineated. Additionally, the heat map plots based on HCA showed additional information about the similarities between samples and clusters [2]

 The variable importance in projection (VIP) scores, which estimate the importance of each variable in the projection used in a PLS model [3] show potential discriminant metabolites with high score values and high discriminator power (Figures 1c and 1d). The annotated metabolites isocoumarin (M301T17), the diterpene lactone (M579T17) (Figure 1c) and the triterpene saponin (M984T24) (Figure 1d) increase in intensity with the progression of the infection, whereas the intensity of the triterpenoid (M685T19), the peonidin 3-(4-sinapoylgentiobioside) (M832T22) (Figure 1c) and the sulfoquinovosyldiacylglycerol (M820T23) (Figure 1d), decrease. The level of annotation and additional information on these metabolites are summarized in supplementary table S1.

Point 3. Several statistical methods are used in this manuscript. For the general audience the rationale of their application has to be described besides the acronym (e.g., VIP scores, PLS-DA) in the section Results.

Response 3. The application and acronym of PCA, PLS-DA and VIP score were described. The bold underlined texts were added.

Line 101: In the first step of the statistical processing, the principal component analysis (PCA), a technique used for  dimensionality reduction of multivariate data whilst preserving most of the variance [1],  was applied to LC-MS metabolic profiles data in order to find cluster of samples.

Line 111: Additionally, the heat map plots based on HCA showed additional information about the similarities between samples and clusters [2]

Line 113: The variable importance in projection (VIP) scores, which estimate the importance of each variable in the projection used in a PLS model [3], show potential discriminant metabolites with high score values and high discriminator power (Figures 1c and 1d).

Line 130: Subsequently, partial least squares-discriminant analysis (PLS-DA) that is a supervised chemometric method used to optimize the separation between different groups [1], was applied to investigate specific metabolic changes in the early asymptomatic infection stage.

Point 4. Since two MS techniques are used in this manuscript I would appreciate the comparison of the two sets of the obtained data using the Venn diagram.

Response 4. The data sets obtained from LC-ESI-MS (Q-TOF) and MALDI-MS (TOF/TOF) are quite different and the comparison through Venn diagram does not provide additional information. LC-ESI-MS can be detecting about 50 times more ions that MALDI-MS. Also, LC-ESI-MS is more comprehensive, being able to detect ions of different polarity while MALDI was focused on lipids using DHB-matrix.

Point 5. An application of MALDI MS to analyze effects of Late blight on tomato decribed by Laurindo et al., 2018 should be mentioned in the Introduction. Results described in [7] should be extensively commented in the context of the data obtained in this report. Interestingly, report by Laurindo et al., 2018 is included in the reference list [7] but still it is not discussed in this context.

Response 5. We mentioned the cited reference in the introduction. The text in bold and underlined was added: “MALDI-MS profiles of proteins, lipids and metabolites have also revealed plant-pathogen interactions [10–12], specifically, the protein profile of sugarcane after infection by Sporisorium scitamineum, [13] the identification of differential proteins of rice leaves infected with the fungus Cochliobolus miyabeanus [14], and the protein that may lead to the resistance of tomato plants to P. infestans [15] (line 70). We don’t fully agree with the reviewer’s opinion that the results from Laurindo 2017 should be extensively discussed along with our data, considering that we investigated the changes in the metabolites expression of infected sensitive plants and not of resistant tomato plants, as is the case of Laurindo. We are currently analyzing unpublished data on the comparison of the onset and development of the infection in sensitive and resistant tomato plants with different species of P. infestans and we totally agree that, in the discussion session of those results, the findings from Laurindo should be extensively commented.

Point 6. Moreover, it would be interesting to see a brief discussion of the metabolic data obtained in this manuscript in light of the transcriptomic data reported by Jiang et al., Plant Cell Rep. 2018, Cui J et al., Theor Appl Genet. 2018 and Cui et al., Plant J. 2017.

Response 6. These references and other related with transcriptomic data of the tomato-P. infestans were introduced in the discussion: 

Line 252: Regarding the specific interaction tomato-P. infestans, tomato resistance to phytopathogen was associated with genes involved with reactive oxygen species (ROS) scavenging systems [10]. Besides, the overexpression of the SpWRKY1 gene in tomato regulates antioxidants to reduce ROS accumulation and alleviate cell membrane injury after P. infestans infection [11,12].

Point 7. Formal comment – I would suggest to replace the term ‘feature’ with ‘signal’ or ‘MS signal’

Response 7: The term feature or molecular feature are more specific to define an MS signal with specific m/z, retention time and adduct, such as [M+H]+ or [M+Na]+ or [M+Cl]-).  These terms are commonly used to refer to metabolites that are not identified or annotated. 

Minor remarks

Point 8. l.29 – ‘late blight’ or ‘Late blight’ - please unify the notation here and through the entire manuscript

Response 8: “late blight” was replaced through the entire manuscript.

Point 9. l.30 – why Phytophthora infestans and Late blight are missing in the key words ?

Response 9: “late blight” and “Phytophthora infestans” keywords were added.

Point 10. l.106 and below – is the accuracy of the m/z values correct ? were the HR measurement performed here ?

Response 10: Yes, the m/z values in brackets correspond to their high-resolution mass.

Point 11. l. 139 – please rewrite the sentence – it is unclear ‘The PLS-DA plots of Figures S8a and S8b verify the closeness between the times of the second cluster.’

Response 11. The sentence was rewritten. “The PLS-DA plots of Figures S8a and S8b confirm the closeness between the post inoculation time points of the cluster II.” (Line 165)

Point 12. l.312 – sphingomyelin is not a phospholipid

Response 12. The word “phospholipids” and “phospholipidic standards” were replaced by “lipids” and “lipidic standards”.

Point 13. l.320 – were plants cultivated in the greenhouse or rather upon the controlled light regime?

Response 13: The plants were cultivated in a permanent photoperiod. The word greenhouse corresponds to the name of the place. The word greenhouse was deleted for clarity.

Point 14. l.322 – delete ‘pure’

Response 14: The word “pure” was deleted.

Point 15. l.324 – ‘V8 media’ – describe the composition or provide the reference

Response 15: The reference (Scanu, et al. 2015) [13] was provided. (Line 357).

Point 16a. l.335 – correct the description: were the entire plants used for the metabolic profiling or just selected leaves?

Response 16a. In the seccion 3.5. The sample preparation was described: The tomato leaves were excised with sterilized scissors and were immediately macerated in liquid nitrogen.

Point 16b. which leaves were collected – only the infected ones?

Response 16b. All the leaves were infected carefully depositing a drop of the sporangial suspension on the leaf with the aid of a micropipette. For the sample preparation all the leaves were collected. The word “all” was added for clarify. “All the tomato leaves were excised with sterilized scissors and were immediately macerated...”

Point 16c. Were any necrotic changes observed?

Response 16c. In the present analysis (96 hours post inoculation) no macroscopically necrotrophic changes were observed.

Point 17. l.341 – replace ‘12,000 g’ with ‘12,000 x g

Response 17: It was done.

Point 18. l.341-4 – delete ‘extract’

Response 18: It was done.

Point 19. l.352 – delete ‘mobile phase’

Response 19: It was done.

Point 20. l.346 – was any external standard used for UPLC-MS ?

Response 20. We didn’t use an external standard for UPLC-MS, but we used QC injections to verify batch and injection influences, besides matrix effect.

Point 21 l.670 – delete ‘61’

 Response 21: It was done.

Point 22. Fig. 1 –a brief explanation of how the VIP plots were generated should be included either in the legend or the main body of the text. Do the color scale presents the intensity of the particular signals – explain in the text. Please explain the Control (mock treatment? Time point ?)

 Response 22. Variable Importance in Projection (VIP), is a weighted sum of squares of the PLS loadings taking into account the amount of explained Y-variation, in each dimension. VIP scores are calculated for each component and were used average of two components to calculate the feature importance. The color scale depends on the range intensity of the metabolites in all samples and the media intensity of the samples the same time. (Line 129)

Point 23. Fig. 2 and 3 – the legend should clearly describe the content of panels c-f, what are the units of Y axis ? formatting of these panels requires correction

 Response 23. We add information about box-plot Y axis in the legend.

In the box plot, Y axis represents intensity of the metabolites as quartile for each sample group related to all data set. The range of the vertical scale is from the minimum to the maximum value of the selected group, or, to the highest or lowest of the displayed reference points, median, and 95% confidence interval of the mean. (Line 155 and 178).

In more details: The top of the rectangle indicates the third quartile, a horizontal line near the middle of the rectangle indicates the median, and the bottom of the rectangle indicates the first quartile. A vertical line extends from the top of the rectangle to indicate the maximum value, and another vertical line extends from the bottom of the rectangle to indicate the minimum value. The illustration shows a generic example of a box plot with the maximum, third quartile, median, first quartile, and minimum values labeled. The relative vertical spacing between the labels reflects the values of the variable in proportion.

Point 24. Fig. 4 – MALDI profiles should be transferred to the supplement

Response 24: The MALDI profiles (Figure 4) were transferred to supplementary information and it was divided in two figures (Figure S11 and S12)

Point 25.  Table 1 – meaning of the ‘Annotation level’ has to explained in the legend. Moreover, comas have to be replaced with dots.

 Response 25: The meaning of the ‘Annotation level’ was explained in the legend of the table 1 and the comas were replaced with dots. The text “Annotation level: identified metabolites (level 1), putatively annotated compounds (level 2), putatively characterized compound classes (level 3), and unknown compounds (level 4)” was added to the table legend  [14,15]. (Line 262). The comas were replaced with dots.

Point 26. Fig.S1 and S2 – legends are not complete – color code (part A) as well as explanation of the meaning of the red circles (part B) have to be provided.

 Response 26. The legend “The red circles correspond to the statistical significance of each molecular feature given by the p-value” was added.

Point 27a. Fig.S3 and S4 – as above –  a brief explanation how the Quality Control was obtained is missing.

Response 27a. The text “The QC samples consisted of a pool of all the different inoculation times and controls and were analyzed at the beginning and at the end of each batch and after every 10 injections” was added to the legend.

Point 27b. Control has to be described too, additionally abbreviation for negative control (CN) is not used in the panels.

Response 27b. The sentenceCoding C_neg means the analysis in negative ionization mode of the controls” was added to define the negative ion mode analyzes of the control samples.

Point 27c. Moreover, notation ‘C_neg’ should be consequently used through the entire manuscript-

Response 27b. The notation C-neg isn't included in the manuscript because it  corresponds to the differentiation of the ionization mode in which the MS analysis was made. However, the control sample are the same.

Point 28. Table S1 – legend - please correct ‘chamical analysis’

Response 28: It was done. It was introduced “chemical analysis”.

1.          Gromski, P.S.; Muhamadali, H.; Ellis, D.I.; Xu, Y.; Correa, E.; Turner, M.L.; Goodacre, R. A tutorial review: Metabolomics and partial least squares-discriminant analysis – a marriage of convenience or a shotgun wedding. Anal. Chim. Acta 2015, 879, 10–23, doi:10.1016/j.aca.2015.02.012.

2.          Ivanisevic, J.; Benton, H.P.; Rinehart, D.; Epstein, A.; Kurczy, M.E.; Boska, M.D.; Gendelman, H.E.; Siuzdak, G. An interactive cluster heat map to visualize and explore multidimensional metabolomic data. Metabolomics 2015, 11, 1029–1034, doi:10.1007/s11306-014-0759-2.

3.          Banerjee, P.; Ghosh, S.; Dutta, M.; Subramani, E.; Khalpada, J.; RoyChoudhury, S.; Chakravarty, B.; Chaudhury, K. Identification of Key Contributory Factors Responsible for Vascular Dysfunction in Idiopathic Recurrent Spontaneous Miscarriage. PLoS One 2013, 8, e80940, doi:10.1371/journal.pone.0080940.

4.          Gupta, R.; Lee, S.E.; Agrawal, G.K.; Rakwal, R.; Park, S.; Wang, Y.; Kim, S.T. Understanding the plant-pathogen interactions in the context of proteomics-generated apoplastic proteins inventory. Front. Plant Sci. 2015, 6, 532, doi:10.3389/fpls.2015.00352.

5.          Ahmad, F.; Babalola, O.O.; Tak, H.I. Potential of MALDI-TOF mass spectrometry as a rapid detection technique in plant pathology: identification of plant-associated microorganisms. Anal. Bioanal. Chem. 2012, 404, 1247–1255, doi:10.1007/s00216-012-6091-7.

6.          Lodha, T.D.; Hembram, P.; Basak, Nitile Tep, J. Proteomics: A Successful Approach to Understand the Molecular Mechanism of Plant-Pathogen Interaction. Am. J. Plant Sci. 2013, 4, 1212–1226, doi:10.4236/ajps.2013.46149.

7.          Que, Y.; Xu, L.; Lin, J.; Ruan, M.; Zhang, M.; Chen, R. Differential Protein Expression in Sugarcane during Sugarcane- Sporisorium scitamineum Interaction Revealed by 2-DE and MALDI-TOF-TOF/MS. Comp. Funct. Genomics 2011, 2011, 1–10, doi:10.1155/2011/989016.

8.          Kim, J.Y.; Wu, J.; Kwon, S.J.; Oh, H.; Lee, S.E.; Kim, S.G.; Wang, Y.; Agrawal, G.K.; Rakwal, R.; Kang, K.Y.; Ahn, I.-P.; Kim, B.-G.; Kim, S.T. Proteomics of rice and Cochliobolus miyabeanus fungal interaction: Insight into proteins at intracellular and extracellular spaces. Proteomics 2014, 14, 2307–2318, doi:10.1002/pmic.201400066.

9.          Laurindo, B.S.; Laurindo, R.D.F.; Fontes, P.P.; Vital, C.E.; Delazari, F.T.; Baracat-Pereira, M.C.; da Silva, D.J.H. Comparative analysis of constitutive proteome between resistant and susceptible tomato genotypes regarding to late blight. Funct. Integr. Genomics 2018, 18, 11–21, doi:10.1007/s10142-017-0570-z.

10.        Cui, J.; Luan, Y.; Jiang, N.; Bao, H.; Meng, J. Comparative transcriptome analysis between resistant and susceptible tomato allows the identification of lncRNA16397 conferring resistance to Phytophthora infestans by co-expressing glutaredoxin. Plant J. 2017, 89, 577–589, doi:10.1111/tpj.13408.

11.        Li, J.; Luan, Y.; Liu, Z. SpWRKY1 mediates resistance to Phytophthora infestans and tolerance to salt and drought stress by modulating reactive oxygen species homeostasis and expression of defense-related genes in tomato. Plant Cell, Tissue Organ Cult. 2015, 123, 67–81, doi:10.1007/s11240-015-0815-2.

12.        Cui, J.; Xu, P.; Meng, J.; Li, J.; Jiang, N.; Luan, Y. Transcriptome signatures of tomato leaf induced by Phytophthora infestans and functional identification of transcription factor SpWRKY3. Theor. Appl. Genet. 2018, 131, 787–800, doi:10.1007/s00122-017-3035-9.

13.        Scanu, B.; Linaldeddu, B.T.; Deidda, A.; Jung, T. Diversity of Phytophthora Species from Declining Mediterranean Maquis Vegetation, including Two New Species, Phytophthora crassamura and P. ornamentata sp. nov. PLoS One 2015, 10, e0143234, doi:10.1371/journal.pone.0143234.

14.        Sumner, L.W.; Amberg, A.; Barrett, D.; Beale, M.H.; Beger, R.; Daykin, C.A.; Fan, T.W.-M.; Fiehn, O.; Goodacre, R.; Griffin, J.L.; Hankemeier, T.; Hardy, N.; Harnly, J.; Higashi, R.; Kopka, J.; Lane, A.N.; Lindon, J.C.; Marriott, P.; Nicholls, A.W.; Reily, M.D.; Thaden, J.J.; Viant, M.R. Proposed minimum reporting standards for chemical analysis. Metabolomics 2007, 3, 211–221, doi:10.1007/s11306-007-0082-2.

15.        Salek, R.M.; Steinbeck, C.; Viant, M.R.; Goodacre, R.; Dunn, W.B. The role of reporting standards for metabolite annotation and identification in metabolomic studies. Gigascience 2013, 2, 13, doi:10.1186/2047-217X-2-13.

Round 2

Reviewer 2 Report

Although not all of my suggestions for amendments were included in the revised manuscript I do accept Authors statements and do not have further comments.